# Cell-free reconstitution reveals centriole cartwheel assembly mechanisms

P. Guichard[1,†,*], V. Hamel[1,†,*], M. Le Guennec[2], N. Banterle[1], I. Iacovache[3], V. Nemčíková[1], I. Flückiger[1], K.N. Goldie[4], H. Stahlberg[4], D. Lévy[5], B. Zuber[3] & P. Gönczy[1]

How cellular organelles assemble is a fundamental question in biology. The centriole organelle organizes around a nine-fold symmetrical cartwheel structure typically ∼100 nm high comprising a stack of rings that each accommodates nine homodimers of SAS-6 proteins. Whether nine-fold symmetrical ring-like assemblies of SAS-6 proteins harbour more peripheral cartwheel elements is unclear. Furthermore, the mechanisms governing ring stacking are not known. Here we develop a cell-free reconstitution system for core cartwheel assembly. Using cryo-electron tomography, we uncover that the *Chlamydomonas reinhardtii* proteins CrSAS-6 and Bld10p together drive assembly of the core cartwheel. Moreover, we discover that CrSAS-6 possesses autonomous properties that ensure self-organized ring stacking. Mathematical fitting of reconstituted cartwheel height distribution suggests a mechanism whereby preferential addition of pairs of SAS-6 rings governs cartwheel growth. In conclusion, we have developed a cell-free reconstitution system that reveals fundamental assembly principles at the root of centriole biogenesis.

[1] Swiss Institute for Experimental Cancer Research (ISREC), School of Life Sciences, Swiss Federal Institute of Technology (EPFL), Lausanne CH-1015, Switzerland. [2] Department of Cell Biology, Sciences III, University of Geneva, Geneva CH-1211, Switzerland. [3] Institute of Anatomy, University of Bern, Bern CH-3012, Switzerland. [4] Center for Cellular Imaging and NanoAnalytics (C-CINA), Biozentrum, University of Basel, Basel CH-4058, Switzerland. [5] Institut Curie, PSL Research University, UMR 168, Centre de Recherche, 26 rue d'ULM, Paris 75231, France. * These authors contributed equally to this work. † Present address: Department of Cell Biology, Sciences III, University of Geneva, Geneva CH-1211, Switzerland. Correspondence and requests for materials should be addressed to P.G. (email: pierre.gonczy@epfl.ch) or to Pa.G. (email: paul.guichard@unige.ch).

The centriole organelle is essential for forming cilia, flagella and centrosomes and is paramount for human health[1–5]. Centrioles are cylindrical structures, typically ~450 nm high and ~250 nm in outer diameter, with a signature nine-fold radially symmetric arrangement of microtubules organized around a cartwheel with likewise symmetry[3,6,7]. The entire cartwheel structure is typically ~100 nm high and located in the proximal part of the centriole, where it is thought to seed organelle formation. The mechanisms directing the assembly of the whole cartwheel remain incompletely understood.

Cryo-tomography of the exceptionally long cartwheel structure in *Trichonympha* sp. revealed the presence of stacked rings that contain a central hub ~22 nm in diameter and that are spaced ~8.5 nm vertically[8]. Nine spokes emanating from the central hub of these rings radiate towards the cartwheel periphery, where spokes originating from two superimposed rings merge, thus yielding ~17 nm vertical spacing. The joined spokes then connect to a pinhead structure that bridges with the peripheral-most microtubule triplets[9].

Each cartwheel ring accommodates nine homodimers of SAS-6 proteins[8], which are required for cartwheel assembly across eukaryotes. For instance, in *Chlamydomonas reinhardtii*, cells lacking the SAS-6 protein Bld12p (hereafter referred to as CrSAS-6) fail to assemble a cartwheel[10]. SAS-6 proteins harbour a globular N-terminal domain followed by an extended coiled coil and an unstructured C-terminal region (Supplementary Fig. 1a). SAS-6 proteins homodimerize via their coiled coil and further oligomerize through interactions between N-terminal domains to form nine-fold symmetrical ring-like assemblies *in vitro*, which are ~4.5 nm thick each[11,12]. Whether such assemblies harbour more peripheral cartwheel elements is not known. Furthermore, the mechanisms enabling individual rings to connect vertically to form the whole cartwheel structure are not known. To tackle these questions, we set out to develop a cell-free reconstitution system of the cartwheel.

## Results

**Native architecture of the core cartwheel *in vivo*.** We first set out to identify the architectural features that must be reconstituted in a cell-free system by determining the native organization of what we term the core cartwheel, namely, the cartwheel without pinhead or microtubules. To this end, we used centrioles purified from *Chlamydomonas*, because examination of previous resin-embedded specimens suggested that the cartwheel structure extends slightly below the region harbouring pinhead and microtubules in this species[13]. Cryo-electron tomography (cryo-ET) allowed us to unveil three regions in the proximal part of the *Chlamydomonas* probasal body (hereafter referred to as the centriole for simplicity), from top to bottom (Fig. 1a–c, Supplementary Fig. 2a, Supplementary Movie 1). First, a region without cartwheel (Fig. 1a). Second, a region in which the cartwheel is present, together with pinhead and microtubules (Fig. 1b,d). Third, a region harbouring the core cartwheel, devoid of pinhead and microtubules (Fig. 1c,e). Moreover, we identified the previously described amorphous ring that surrounds the cartwheel[14], as well as recruitment of the first microtubule of the triplet (A-microtubule) to the core cartwheel (Supplementary Fig. 2b–e). Measurements showed that the native core cartwheel comprises a hub ~22 nm in diameter, from which emanate nine spokes extending until two peripheral densities that we termed D1 and D2, which are located, respectively, at ~37 and ~46 nm from the hub margin (Fig. 1e,f). These architectural features define the organization of the core cartwheel that a cell-free assay ought to reconstitute.

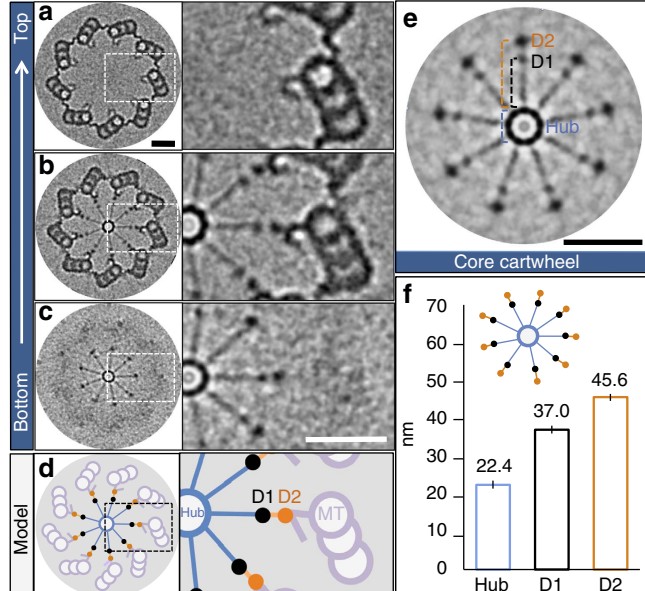

**Figure 1 | Native architecture of the core cartwheel *in vivo*.** (**a–c**) Images from cryo-tomogram of purified *Chlamydomonas* centriole (probasal body) after nine-fold symmetrization (left, three regions from top to bottom of centriole), with corresponding insets of boxed regions (right). (**d**) Schematic of centriole in **b**, with corresponding inset. (**e**) Native architecture of core cartwheel after symmetrization, highlighting the hub, as well as D1 and D2 densities. (**f**) Measurements of centriole features (see **e**): hub diameter (22.4 ± 1.3 nm, n = 10), distances from the hub margin to D1 (37 ± 1 nm, n = 13) and D2 (45.6 ± 0.8 nm, n = 13). Data extracted from five tomograms. Scale bars correspond to 50 nm.

**Core cartwheel assembly *in vitro*.** We set out to address whether CrSAS-6 could assemble not only the hub, as previously established[11,15], but also D1 and D2. To this end, we developed a cell-free assay enabling efficient formation of cartwheel-like assemblies. In brief, soluble recombinant proteins were dialysed against 10 mM K-PIPES pH 7.2 and the resulting structures formed in solution then analysed by cryo-electron microscopy (cryo-EM) (Methods). As shown in Fig. 2a–c and Supplementary Fig. 3a, we found that full-length CrSAS-6 (CrSAS-6_FL, Supplementary Fig. 1a,c) formed dense structures containing numerous rings ~20.5 nm in diameter. However, spokes were not well organized; moreover, D1 and D2 were not detectable (Fig. 2b). Therefore, CrSAS-6_FL cannot alone form the core cartwheel under these assay conditions.

We hypothesized that this inability might stem from a requirement of a CrSAS-6 interacting partner to assist in core cartwheel formation. Bld10p appeared interesting in this respect, because it is essential for cartwheel formation in *Chlamydomonas*[16]. Moreover, using structured illumination microscopy, we found that both CrSAS-6 and Bld10p localized at the cartwheel (Fig. 2d–f), in line with previous immuno-electron microscopic data[10,16]. Intriguingly, the Bld10p human homologue Cep135 interacts through its C-terminal region with HsSAS-6 (ref. 17), whereas a *Chlamydomonas* mutant lacking this region exhibits defective centriole assembly[16]. To test the possibility that the C-terminus of Bld10 interacts with CrSAS-6_FL and thus promotes core cartwheel assembly, we performed the cell-free assay with CrSAS-6_FL plus a fragment encompassing residues 1061–1641 of Bld10p (Bld10p_C; Supplementary Fig. 1b,d). This led to the formation of large assemblies containing the two proteins (Supplementary Fig. 3b,c). Importantly, analysis by cryo-EM revealed regular

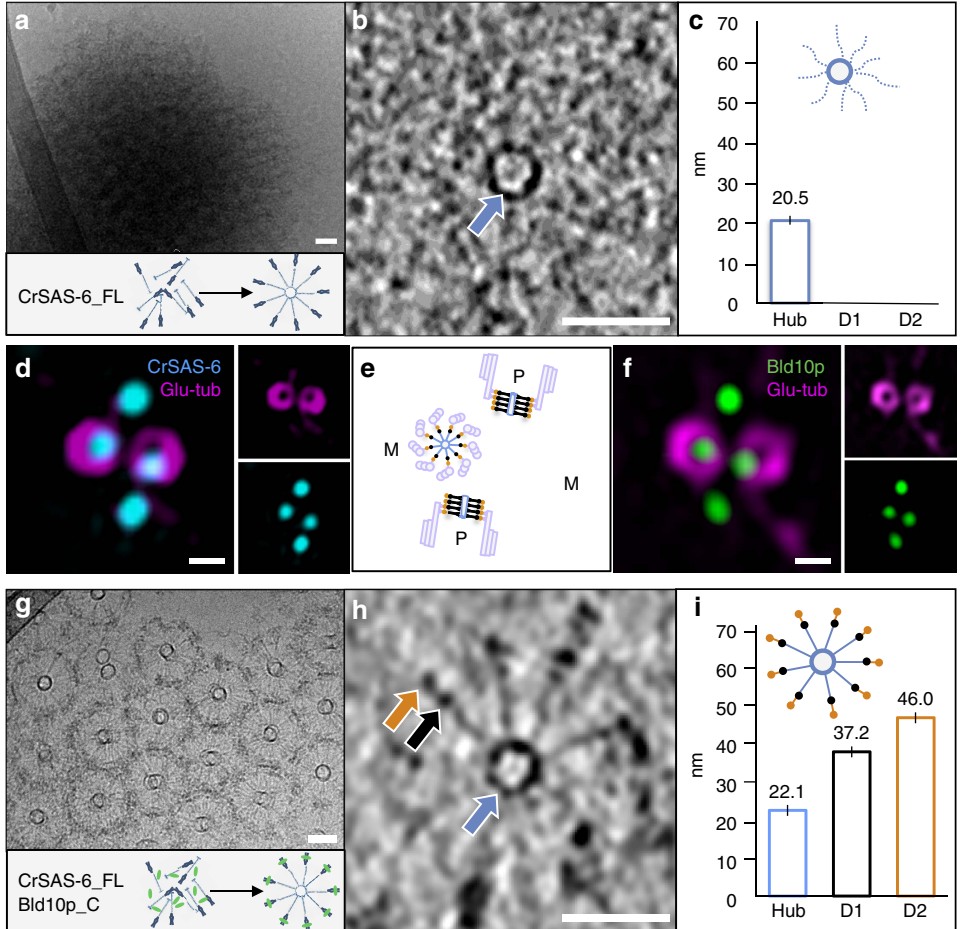

**Figure 2 | CrSAS-6 and Bld10p together mediate core cartwheel formation *in vitro*.** (**a**) Cryo-EM of CrSAS-6_FL assemblies. (**b**) High-magnification cryo-ET of CrSAS-6_FL assemblies, highlighting the hub (blue arrow). (**c**) Measurements of hub diameter (20.5 ± 1.8 nm, *n* = 50, data extracted from two tomograms). (**d**–**f**) Structured illumination microscopy of purified *Chlamydomonas* centrioles highlighting localization of CrSAS-6 (**d**, cyan) and Bld10p (**f**, green) at the proximal end, together with corresponding schematic (**e**). Glu-tub: glutamylated tubulin (magenta), marking mature centrioles. Scale bar: 250 nm. (**g**) Cryo-EM of CrSAS-6_FL/Bld10p_C arrays. (**h**) High-magnification cryo-ET of CrSAS-6_FL/Bld10p_C arrays, highlighting hub (blue arrow), D1 (black arrow) and D2 (orange arrow). (**i**) Measurements of hub diameter (22.1 ± 1.4 nm, *n* = 9), distances from the hub margin to D1 (37 ± 0.9 nm, *n* = 12) and D2 (46 ± 0.8 nm, *n* = 12). Data extracted from two tomograms. Scale bars correspond to 50 nm.

honeycomb-like arrays resembling three-dimensional (3D) organized cartwheels (Fig. 2g and Supplementary Fig. 3d). Corresponding cryo-tomograms established that these arrays displayed the characteristic features of the core cartwheel (Fig. 2h and Supplementary Fig. 3e,f). Thus these arrays often harboured a hub ~22 nm in diameter, from which emanated spokes terminated by two densities located, respectively, at ~37 and ~46 nm from the hub margin (Fig. 2i), matching the locations of D1 and D2 *in vivo* (Fig. 1f). We conclude that, together with the C-terminus of Bld10p, CrSAS-6_FL can form assemblies resembling the core cartwheel.

**CrSAS-6 enables cartwheel stacking.** The above findings taken together raise the possibility that the C-terminal region of CrSAS-6_FL, when not in a complex with Bld10p, negatively regulates cartwheel assembly. We tested whether removing this region leads to the autonomous assembly of cartwheel-like structures. Remarkably, we discovered that a C-terminal truncation of CrSAS-6, CrSAS-6_NL (Supplementary Fig. 1e) indeed efficiently assembled a 3D lattice with interconnected cartwheel-like units (Fig. 3a–c, Supplementary Movie 2). These units exhibited eight- and nine-fold symmetries (60% and 40%,

respectively, *n* = 207; Fig. 3c, Supplementary Fig. 4a), compatible with those obtained with recombinant SAS-6 proteins[11,12,18]. We determined that the lattice units terminated ~37 nm away from the hub margin, corresponding to the position of D1, and were lacking D2 (Fig. 3c,d). The CrSAS-6 coiled coil is present in its entirety in CrSAS-6_NL and is predicted to be ~45 nm long[9] (Supplementary Fig. 1a), raising the possibility that it adopts a non-elongated conformation towards the periphery, where D1 is located. The D1 densities within the lattice were most discernable with cryo-EM following circularization (Fig. 3e), as well as with negative staining EM of individual CrSAS-6_NL assemblies, which confirmed its presence ~37 nm away from the hub margin (Fig. 3f,g, Supplementary Fig. 4b). These observations demonstrate that the D1 density is composed primarily of CrSAS-6 itself.

Together, these experiments reveal that the C-terminal region of CrSAS-6_FL exerts a negative regulation that is abrogated through interaction with the C-terminus of Bld10p. Moreover, they demonstrate that CrSAS-6_NL possesses an autonomous ability to assemble organized lattices of cartwheel-like structures. The fact that these lattices are more organized than the CrSAS-6_FL/Bld10p_C assemblies may be due to the fact that this fragment of Bld10p does not

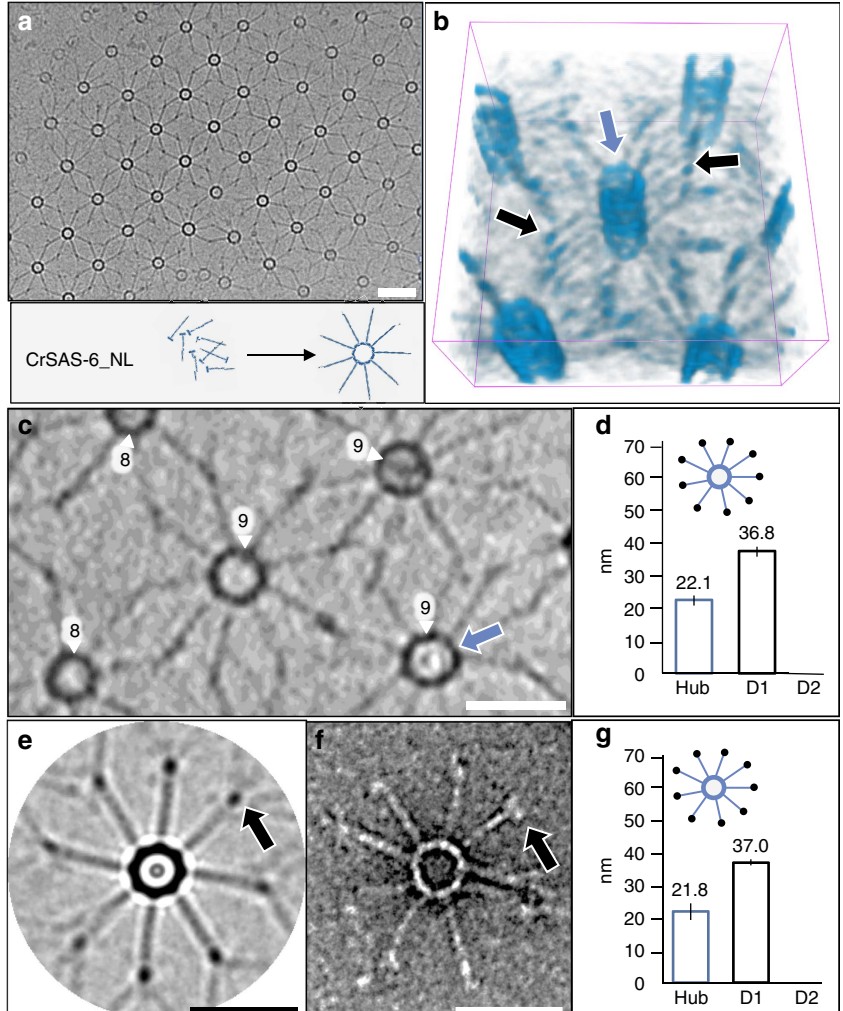

**Figure 3 | Intrinsic properties of CrSAS-6 enable cartwheel ring stacking.** (**a**) Cryo-EM of CrSAS-6_NL lattice. (**b**) 3D representation of CrSAS-6_NL cartwheels within the lattice. Note stacked rings (blue arrow) and spokes in register (black arrows). (**c**) Cryo-ET section of CrSAS-6_NL lattice, highlighting eight- and nine-fold symmetrical structures, with hub (blue arrow). (**d**) Measurements in cryo-ET images of hub diameter (22.1 ± 1.3 nm, *n* = 50, only nine-fold symmetrical rings, data extracted from four tomograms) and distance from hub margin to the end of the coiled coil (36.8 ± 1.2 nm, *n* = 28). (**e,f**) The C-terminal extremity of CrSAS-6_NL forms a density visible in cryo-EM after circularization of 8-fold symmetrical cartwheels (**e**) or in negative stain EM (**f**). (**g**) Measurements in negative stain EM images of hub diameter (21.8 ± 2.5 nm, *n* = 32, one experiment) and distance from hub margin to the terminal density (37 ± 0.9 nm, *n* = 34, one experiment). Scale bars correspond to 50 nm.

completely relieve the inhibitory effect of the C-terminus moiety of CrSAS-6.

We next investigated whether 3D cartwheel-like structures reconstituted *in vitro* exhibited the characteristic vertical stacking present *in vivo* in *Trichonympha* sp. (Supplementary Fig. 5a)[9]. Remarkably, subtomogram averaging demonstrated that spokes emanating from CrSAS-6_NL rings were spaced ∼8 nm vertically (Fig. 4a,b) and that spokes merged two by two towards the periphery (Fig. 4a, arrowheads), as observed *in vivo*[9]. Furthermore, analysis of cryo-tomograms established that the vertical periodicity of these peripheral elements was ∼17 nm (Fig. 4c and Supplementary Figs 5b and 6), again similar to the *in vivo* situation. CrSAS-6_FL/Bld10p_C structures exhibited analogous vertical spoke periodicities of ∼8 nm at the hub margin (Supplementary Fig. 5c blue, Supplementary Fig. 5d) and of ∼15 nm towards the periphery (Supplementary Fig. 5c green, Supplementary Fig. 5e). Overall, these findings demonstrate that CrSAS-6_NL alone, as well as CrSAS-6_FL/Bld10p_C, form stacked cartwheels *in vitro* with a vertical organization akin to that observed *in vivo*.

**Mechanism of cartwheel assembly.** We set out to investigate the mechanisms underlying vertical growth of reconstituted cartwheel structures. We reasoned that analysing cartwheel height distribution could provide critical clues. We focussed on CrSAS-6_NL because the corresponding lattice was particularly well organized, although analogous conclusions were reached with CrSAS-6_FL/Bld10p_C (Supplementary Fig. 3g). We measured the height of 1,039 cartwheels assembled by CrSAS-6_NL (Fig. 4d and Supplementary Movies 3–5). Importantly, this revealed that the mean height of reconstituted cartwheels was ∼107 nm, similar to that observed in *Chlamydomonas* cells[19]. Moreover, although the raw data of cartwheel height distribution were noisy (Supplementary Fig. 7), plotting it using a running average revealed peaks in cartwheel height frequencies that appeared to be spaced every ∼17 nm for structures smaller than ∼134 nm and every ∼8 nm for those taller than that (Fig. 4d,e).

We sought to develop a simple model of cartwheel height distribution reproducing the apparent peaks in height frequencies and thus uncover underlying assembly mechanisms.

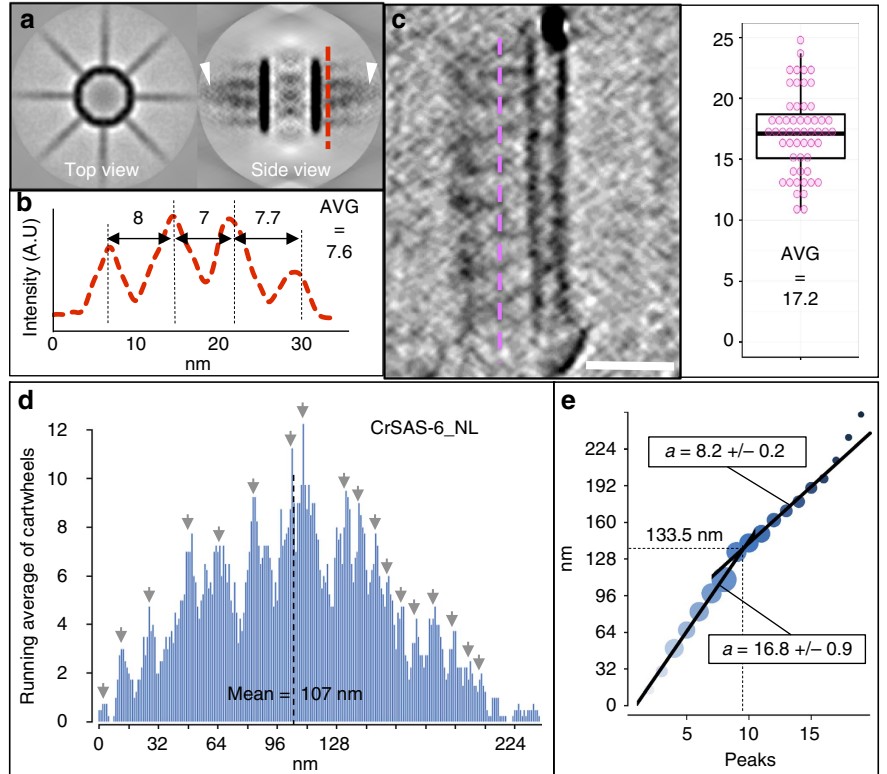

**Figure 4 | Intrinsic properties of CrSAS-6 enable cartwheel ring stacking.** (**a**) Average of 1,668 subtomograms of reconstituted eight-fold symmetrical cartwheels (top and side views). Note radial spokes merging two by two (side view, arrowheads). (**b**) Plot profile across average subtomogram side view (at red dashed line in **a**), showing average 7.6 ± 0.5 nm periodicity of spokes close to the hub. (**c**) Cryo-ET of CrSAS-6_NL highlighting peripheral periodicity of spokes; the purple dashed line illustrates the region from which the quantification of periodicities shown on the right was extracted (n = 55 peripheral elements, data extracted from two tomograms). Scale bars to correspond to 50 nm. (**d**) Experimental height distribution of CrSAS-6_NL reconstituted cartwheels smoothed with a running average of four (light blue); dotted line shows the mean (107 ± 48 nm, n = 1,039; data extracted from four tomograms). (**e**) Regression lines using distribution peaks ranging from 4 to 133.5 nm (a = 16.8 ± 0.9) and above (a = 8.2 ± 0.2). Peaks used are marked by grey arrows in **d**. Weighted dots represent the number of cartwheel at each peak.

We reasoned that if cartwheel growth occurred through the sequential addition of single SAS-6 rings, then cartwheel height should increase by ∼8.5 nm at each incorporation step, resulting in a height distribution with peaks at multiples of ∼8.5 nm. We further considered that double SAS-6 rings could be the basic growth unit, either because they assemble in solution or because the incorporation of a single ring triggers the immediate addition of a pairing one. Regardless, the height distribution should display peaks at multiples of ∼17 nm if only double rings are incorporated.

Given the above considerations, we developed a mathematical fitting model for the experimentally measured distribution (see Methods section). In brief, the model consists of an overall distribution that results from two independent contributions, one from a height distribution of single rings and the other from a height distribution of double rings. The relative contribution of these two species to average cartwheel height is expressed with the variable $R$, whereby $R = 0$ corresponds to single rings only and $R = 1$ to double rings only. Moreover, we introduced a free offset parameter $s$, which could reflect, for instance, a different height increase accrued by the first or the last ring within the stack. We set out to determine which set of model parameter values best fitted the experimental data using the Akaike Information Criterion (AIC). We varied the inter ring spacing $h_1$ accrued following addition of a single ring from 3 to 11 nm and the inter ring spacing $h_2$ gained after addition of a double ring from 12 to 20 nm. Additionally, the variable

$R$ was changed from 0 to 1, and the free offset parameter $s$ from 0 to 4 nm.

As shown in Fig. 5a, this analysis demonstrated that, for single rings, the inter ring spacing giving rise to the lowest AIC value was of ∼8 nm, although divergence from this value did not impact the goodness of fit in a dramatic manner, whereas it was contained in a narrow range with an optimum at ∼17.5 nm for double rings. Remarkably, these values are close to those determined by measuring spacing directly (see Fig. 4a–d). Moreover, the lowest AIC value was obtained with the mixed addition of single and double rings, with the latter contributing ∼85% of cartwheel height on average (Fig. 5b, $R = 0.85$). Furthermore, we observed that the lowest AIC value was obtained with an offset $s$ of ∼3 nm (Fig. 5b), although having an offset in the range of ∼2–4 nm only mildly affected the goodness of fit. The finding of such an offset is intriguing because we noted that, whereas a few cartwheel ∼4 nm in height were observed experimentally, no cartwheel ∼8 nm in height was found (see Fig. 4d). Although numbers of such small cartwheels are too low to ascertain whether this observation is significant, the offset suggested by the modelling may reflect the fact that the assembly process begins with a single ring ∼4 nm in height.

Overall, while further analysis is needed to understand cartwheel assembly mechanisms in full, the effective mathematical model developed here captures some of the salient features of the experimental cartwheel height distribution. Of particular importance, the model establishes that the experimentally

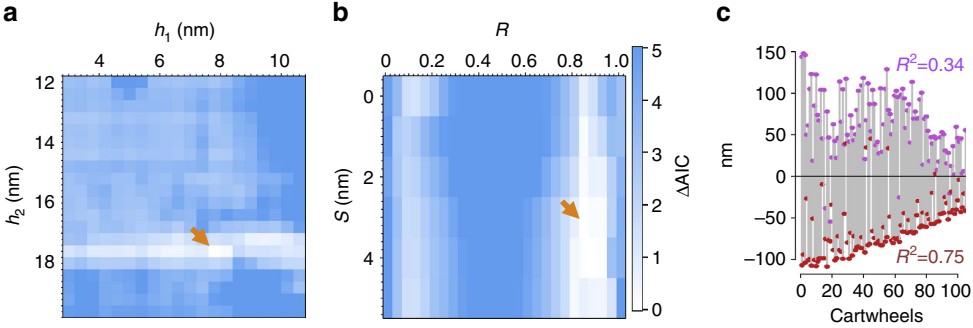

**Figure 5 | Stepwise assembly of *in vitro* reconstituted cartwheels. (a)** AIC differences (ΔAICs) as a function of the inter ring spacing for single ring ($h_1$) and double rings ($h_2$) for $R = 0.85$ and $s = 3$ (see **b**) showing that the heights minimizing the ΔAICs are $h_1 = 7.8$ and $h_1 = 17.6$ (orange arrow), with a narrower range for $h_2$ than for $h_1$. **(b)** The ΔAICs corresponding to the ($h_1,h_2$) pairs minimizing the AIC computed at varying offsets ($s$) and abundance ratios ($R$), showing that the best fitting model is characterized by $R = 0.85$ and $s = 3$ (orange arrow). Note that whereas the local minimum at $R = 0.85$ is well defined, the ΔAIC displays a weaker dependence on the value of $s$. **(c)** Distances from a global regression plane (central black line) of the top and bottom extremities of individual cartwheels within a tomogram ($n = 105$ cartwheels). Regression planes for the two extremities were computed and corresponding $R^2$ values were calculated (lower $R^2 = 0.34$, violet; higher $R^2 = 0.58$, red). The x axis represents individual cartwheels.

observed distribution is most accurately recapitulated when considering the addition of both single and double rings of SAS-6, with recruitment of pairs of SAS-6 rings being the preferred mode contributing to cartwheel size increase.

We next investigated whether vertical cartwheel growth might occur anisotropically, with preferential subunit addition of rings on one side, or instead in an isotropic manner. We reasoned that anisotropic growth should yield more variability on the side of preferred rings addition, whereas isotropic growth is anticipated to result in analogous variability on the two sides. Therefore, we calculated linear regressions of the cartwheels' extremity positions and determined the proportion of variance ($R^2$) explained by the regression for each side of the lattice (Fig. 5c, Supplementary Fig. 8a and Supplementary Movie 6). This revealed that the two extremities exhibited different variability, with one flat and near-homogenous end (Fig. 5c, red) and one more heterogeneous one (Fig. 5c, violet), without correlation with lattice orientation on the EM grid (Supplementary Fig. 8b). These findings support the notion that ring addition occurs preferentially on one side of the growing structure, thus ensuring polarized cartwheel assembly.

## Discussion

Overall, our work suggests a mechanism of cartwheel stacking relying on the preferential polarized incorporation of pairs of SAS-6 rings onto a single seed ring (Fig. 6). How could such stacking be achieved? Several mechanisms could be envisaged. For instance, stacking could rely on D1 densities, which comprise the end of the CrSAS-6 coiled coil and could serve as pillars connecting successive rings. Alternatively, spokes themselves could be oriented either downwards or upwards, so that some could connect with the ring just below and others with the ring just above, thus ensuring vertical continuity. A third scenario stems from the presence of a density within the hub of CrSAS-6_NL structures (see Fig. 4a, side view), which could reflect the unusual folding of part of the CrSAS-6 coiled coil and serve as a vertical connecting device. Regardless of the mechanism, our analysis suggests that cartwheel growth occurs preferentially in a polarized manner, perhaps because one side of the growing cartwheel is somehow primed for the further addition of SAS-6 rings (Fig. 6).

What features of SAS-6 proteins determine cartwheel height? Length regulation mechanisms in other systems include ones

relying on a molecular ruler, on a limiting pool of precursor proteins or else on a dynamic equilibrium between assembly and disassembly reactions[20]. Which of these mechanisms, if any, applies in the case of the cartwheel requires further investigation, including in systems such as *Trichonympha* sp. where the cartwheel reaches exceptional heights[21]. Moreover, it will be important to complement the reconstitution system introduced here with live assays of cartwheel assembly to uncover the relevant kinetic parameters and thus achieve a fuller understanding of the underlying mechanisms.

What drives the assembly of the core cartwheel? We discovered that full-length CrSAS-6 together with a fragment of Bld10p is sufficient to assemble structures *in vitro* that resemble the core cartwheel found *in vivo*. Intriguingly, such assemblies also bear resemblance to the tubules generated in *Drosophila* spermatocytes upon combined overexpression of DmSAS-6 and Ana2/STIL[22]. Therefore, SAS-6 proteins possess the ability to generate stacked entities in different systems, with Bld10p and/or Ana2/STIL playing an important role as associated components.

In conclusion, we have developed a novel cell-free assay that allowed us to reconstitute the core cartwheel structure and unravel fundamental assembly mechanisms, thus providing a critical entry point towards unravelling the building principles of the entire centriole organelle.

## Methods

**Cloning of GST-HisCrSAS-6_FL and GST-Bld10p_C.** The GST-HisCrSAS-6_FL construct was prepared by introducing a 6xHis tag between the CrSAS-6 sequence and the GST-tag in the pGEX-CrSAS-6_FL clone[23]. Briefly, the following DNA sequence of *Chlamydomonas* CrSAS-6 (1–876 bp) flanked by the restriction sites BamHI and SacI was PCR-amplified from pGEX-CrSAS-6_FL using the following primers: 6HisCrSAS-6-F: 5′-GGATCCCACCATCATCATCATCATAT GCCGCTTCTTCTC-3′ and SacICrSAS-6-R(871): 5′-GAGCTCCGACACCT TGGTGTCCAG-3′. The PCR fragment was subcloned in a pGEMT vector (Promega) prior to sequence verification. Thereafter, the DNA encoding the modified fragment of 6His-CrSAS-6_FL was digested by BamHI and SacI and placed back into the original pGEX-CrSAS-6_FL vector. The final construct allowed the production of a fusion protein consisting of GST, a Prescission Protease recognition sequence, followed by a 6xHis tag and the CrSAS-6_FL sequence. Note that this construct has been generated because GST-CrSAS-6_FL was insoluble and because the His-tagged version of CrSAS-6-FL gave rise mainly to truncated products when expressed in *Escherichia coli*.

The DNA fragment coding for Bld10p_C (residues 1,061–1,641) was PCR-amplified from a full-length *Chlamydomonas* Bld10p clone (gift from Masafumi Hirono)[16] using the following primers: SpeI-bld10(1061)F: 5′-GCACT AGTAGTGCGGAGCAGCTGCTA-3′ and SpeI-bld10-stop-R: 5′-ATACTAGTCT ACCGACGCGGCCC-3′ and subcloned in a pGEMT vector. After sequence

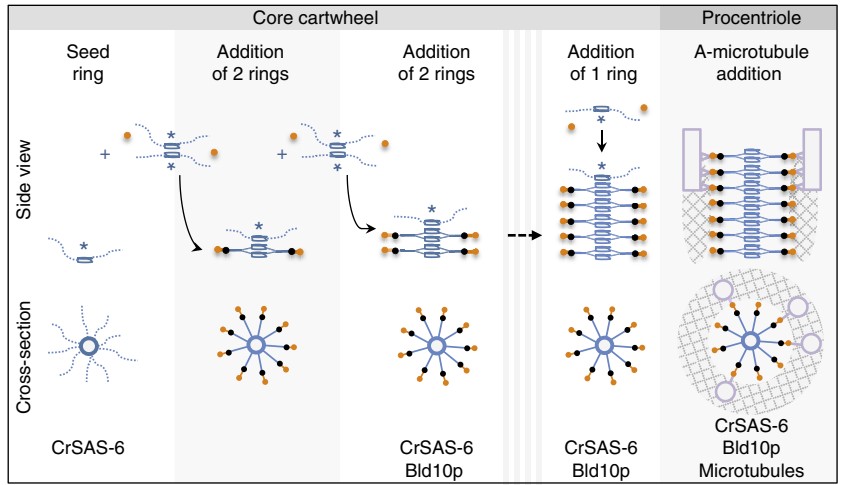

**Figure 6 | Working model of cartwheel assembly.** A single ring of SAS-6 (blue) serves as an initial seed onto which pairs of SAS-6 rings are recruited preferentially in a stepwise manner. Our work suggests that the D1 density (black) could arise from the C-terminus of the CrSAS-6 coiled coil, whereas the D2 density appears upon interaction with a fragment of Bld10p (orange). The amorphous ring and the initial microtubules recruited to the core cartwheel are represented as well; note that the amorphous ring could be present *in vivo* earlier than represented. Asterisks represent the capability of a SAS-6 ring to assemble with another one. In one hypothetical scenario, cartwheel growth could be terminated by the addition of a single SAS-6 ring, which, in contrast to the addition of double rings, would not endow the cartwheel with the capacity to further incorporate incoming SAS-6 rings. See text for further details.

verification, the amplified DNA fragment was transferred into a modified pGEX-6P vector using the SpeI restriction site.

**Protein expression and purification.** The *E. coli* strain BL21(DE3) (Stratagene) was used for protein expression. Bacteria were grown at 37 °C to an $OD600_{nm}$ of 0.5 in LB medium containing the appropriate antibiotics (ampicillin for GST-HisCrSAS-6_FL and GST-Bld10p_C, kanamycin for His-Stag-CrSAS-6_NL(1-503) and 6xHis-CrSAS-6_NL(1-503))[11]. The temperature was then switched to 20 °C and the culture induced by the addition of 0.5 mM IPTG for 16 h.

Bacterial cells were collected by centrifugation at 4,000*g* for 10 min (JLA-9.1000, Beckman Coulter) and the pellet was washed with PBS. Lysis was achieved by resuspending the bacterial pellet in 50 mM Tris pH 7.5, 500 mM NaCl supplemented with proteases inhibitors (Complete EDTA-free, Roche) and incubated with lysozyme (1.6 ml of 15 mg ml$^{-1}$ stock solution) for 10 min on ice prior to sonication. Finally, 0.5% Triton X-100 was added to the cell lysate, before spinning 30 min at 12,000*g* at 4 °C (JA 25.50, Beckman Coulter).

For purification of His-Stag-CrSAS-6_NL, 20 mM imidazole was added to the soluble lysate before incubation with Ni-NTA beads for 1 h at 4 °C on a rotating wheel. Ni-NTA beads were then washed once with 50 mM Tris pH 7.5, 800 mM NaCl, 0.5% Triton X-100, 40 mM imidazole and protease inhibitors, and twice with lysis buffer. Elution of His-Stag-CrSAS-6_NL was performed with 50 mM Tris pH7.5, 500 mM NaCl and 400 mM imidazole. The eluted material was then subjected to size exclusion chromatography using a HiLoad Superdex 200 16/600 PG (GE Healthcare, 28989335) in 50 mM Tris pH 7.5 and 150 mM NaCl. His-Stag-CrSAS-6_NL proteins were concentrated to 0.89 mg ml$^{-1}$ in 50 mM Tris pH 7.5 and 150 mM NaCl.

For purification of CrSAS-6_NL protein[11] for negative staining EM, the recombinant protein was purified by immobilized metal-affinity chromatography on HisTrap HP Ni2 + -Sepharose columns (GE Healthcare) at 4 °C according to the manufacturer's instructions. The 6xHis tag was cleaved during dialysis against thrombin cleavage buffer (20 mM Tris-HCl, pH 7.4 supplemented with 150 mM NaCl and 2.5 mM CaCl2) for 16 h at 4 °C using 2 units of human thrombin (Sigma) per milligram of recombinant protein. Cleaved samples were reapplied to immobilized metal-affinity chromatography to separate the cleaved products from the 6xHis-tagged protein, concentrated and gel-filtrated on a SEC HiLoad Superdex 200 16/60 column (GE Healthcare) equilibrated in 20 mM Tris-HCl pH 7.5 supplemented with 150 mM NaCl and 2 mM dithiothreitol (DTT). All purified proteins were snap-frozen in liquid nitrogen and stored at − 80 °C.

For purification of GST-Bld10p_C, the soluble lysate was incubated with glutathione agarose beads (GE Healthcare) for 1–2 h at 4 °C on a rotating wheel. Beads were then washed twice with lysis buffer with 1% Triton X100 and three times with cleavage buffer (20 mM Tris pH 7.5, 150 mM NaCl, 0.5 mM EDTA, 1 mM DTT). Elution of Bld10p_C was achieved by incubating beads with Prescission Protease (GE Healthcare) overnight at 4 °C in cleavage buffer. Buffer exchange (10 mM K-PIPES, 150 mM NaCl, 1 mM DTT) and concentration of cleaved Bld10p_C was performed using an Amicon ultra 10 K centrifugal device to a final concentration of 5 mg ml$^{-1}$.

For purification of GST-HisCrSAS-6_FL, the soluble lysate was also incubated with glutathione agarose beads for 1–2 h at 4 °C on a rotating wheel. Beads were then washed twice with 50 mM Tris pH 7.5, 1 M NaCl, 1% Triton X100, 1 mM DTT and 5 mM EDTA, then once in lysis buffer and finally once in cleavage buffer (20 mM Tris pH 7.5, 150 mM NaCl, 0.5 mM EDTA, 1 mM DTT) for 10 min at 4 °C. Elution of HisCrSAS-6_FL was achieved by incubating beads with Prescission Protease overnight at 4 °C in cleavage buffer to remove GST. HisCrSAS-6_FL at ∼0.8 mg ml$^{-1}$ was snap frozen directly after cleavage in the cleavage buffer to prevent precipitation during dialysis.

**CrSAS-6-based cartwheel assembly *in vitro*.** Five microlitres of soluble proteins, either CrSAS-6_FL (10 μM) or His-Stag-CrSAS-6_NL (20 μM) that had been purified in 20 mM Tris pH 7.5 and 150 mM NaCl, were dialysed for >16 h against 2 litres of 10 mM K-PIPES pH 7.2 using the slide-A-lyzer mini dialysis unit (3.5 K, 10–100 μl, reference 69,550, Pierce). Approximately ∼100 μl of dialysed proteins were recovered the next day (final molarity for CrSAS-6_FL ∼0.5 μM, for CrSAS-6_NL ∼1 μM), 5 μl of which were further processed for cryo-EM.

For CrSAS-6_FL (10 μM)/Bld10p_C (78 μM) assemblies, 5 μl of each protein were first mixed and incubated for 4 h at 4 °C in 20 mM Tris pH 7.5, 150 mM NaCl prior to dialysis and analysis as delineated above (final molarity for CrSAS-6_FL ∼0.5 μM, for Bld10p_C ∼4 μM).

**EM-negative staining.** Protein sample (CrSAS-6 NL) was diluted in 10 mM K-PIPES pH 7.0 to a final concentration of 3 μg ml$^{-1}$ and deposited on glow-discharged carbon-coated 400 mesh copper grids (Canemco Inc.) for 2 min. Excess protein was washed with 10 mM K-PIPES pH 7.0. The sample was then stained with 2% uranyl acetate, blotted with a filter paper (Whatman no. 1) to remove excess stain and air dried.

Electron micrographs were taken in a Tecnai Spirit (FEI) operated at 120 kV equipped with an Eagle CCD (charge-coupled device) camera (FEI).

**Electron cryo-ET of *in vitro* reconstructed cartwheel.** Cartwheels reconstituted with His-Stag-CrSAS-6_NL were diluted with 10 nm fiducial gold beads (AURION, BSA tracer ref. 210.133) in 10 mM K-PIPES pH 7.2 and then deposited onto a Lacey carbon film grid (300 Mesh, EMS), blotted on the opposite side with a filter paper (Whatman no. 1) and vitrified in liquid ethane with a homemade plunging system.

Tomograms used to generate images shown in the manuscript were recorded on a cryo-electron microscope JEOL JEM 2200FS operating at 200 kV, equipped with a field emission gun and a Ω filter. Images were collected using a 2k × 2k CCD camera (Gatan, Pleasanton, CA, USA) with a single axis Z-loss tomographic tilt series at 4–6 μm defocus acquired from − 55° to 55° at 2° increment angles. Nominal magnification and energy window were × 20,000 (0.54 nm pixel size) and 20 eV, respectively.

Tomograms used for height measurements and subtomogram averaging were recorded on a Tecnai F20 field emission gun electron microscope (FEI Eindhoven,

The Netherlands) operating at 200 kV. Tilt series covering an angular range from $-60°$ to $+60°$ in $1°$ increments were recorded at $\times 29,000$ using an $4K \times 4K$ Falcon 2 direct electron detector (FEI, 0.35 nm final pixel size, $-2,5\,\mu m$ defocus) with the FEI Tomography Software 4.0.

All tomograms were aligned using the 10 nm fiducial gold beads and reconstructed by R-weighted back projection with IMOD[24]. Note that we carefully analysed the tomograms to ensure that the vitreous ice was thicker than the cartwheel lattice to prevent artefacts of the air–water interface.

For subtomogram averaging, 1,668 subtomograms were extracted from 4 independent tomograms as 200 pixel $\times$ 200 pixel $\times$ 80 pixel volumes. Volumes were prealigned with SPIDER[25], using as reference the 3D map of the *Trichonympha* sp. cartwheel[9] The prealigned boxes were subsequently re-aligned with DYNAMO[26], using the same reference. An average was performed on the aligned boxes, and the average symmetrized to obtain a faithful compound structure of the cartwheel. Note that subtomogram averaging has been performed on the eight-fold cartwheels due to their higher homogeneity of spokes' organization. Note also that the low-resolution longitudinal view inherent to the missing wedge in tomography and the invariant top view orientations did not allow high-resolution reconstruction to be conducted. The final resolution determined using Fourier Shell Correlation was 60.6 Å.

For cartwheel height measurements, the top and bottom of a cartwheel were detected manually in a volume. The Euclidian distance was computed between the $x$, $y$ and $z$ coordinates of these two points. The obtained value was then multiplied by the pixel size of the acquired tomogram.

For measurements of distances from the hub margin to D1 or D2 densities, only spokes terminated by a D1 and/or a D2 density were measured.

All images have been processed using ImageJ[27] and the 3D rendering has been generated using UCSF Chimera[28].

### Isolation and cryo-ET of *Chlamydomonas* centrioles.

The cell-wall-less *Chlamydomonas* strain CW-15 was grown in Tris-acetate-phosphate (TAP medium containing Trace)[29]. *Chlamydomonas* centrioles were isolated as described[9]. Briefly, cells were pelleted at 300g in an Eppendorf 5810R centrifuge, washed three times with TAP buffer and resuspended for 10 min in 10 ml of K-PIPES 10 mM supplemented with anti-protease cocktail (Roche, complete EDTA-free), NP40 1% and 0.01 M EDTA. After centrifugation at 600g in an Eppendorf 5810R centrifuge to remove cell debris, the supernatant was centrifuged in a 15 ml Corex tube (JS13.1, Beckman Coulter) at 2,000g during 10 min into 1 ml of a 60% sucrose cushion. The 2 ml cushion interface (1 ml cushion + 1 ml above the cushion) was diluted 1:1 with 10 mM K-PIPES pH 7.2. The resulting suspension of nucleus–centriole complex was stirred by pipetting to detach the nucleus and then centrifuged a 1,000g on a tabletop centrifuge to pellet the nucleus. The resulting suspension was again diluted in 10 ml of 10 mM K-PIPES pH 7.2.

*Chlamydomonas* centrioles were analysed as described[9]. In brief, purified *Chlamydomonas* centrioles were analysed by cryo-ET with a cryo-electron microscope JEOL JEM 2200FS operating at 200 kV, equipped with a field emission gun and a Ω filter. Images were collected using a 2k × 2k CCD camera (Gatan) with a single axis Z-loss tomographic tilt series at 4–6 µm defocus acquired from $-55°$ to $55°$ at $1°$ increment angles. Nominal magnification and energy window were $\times 15,000$ (0.74 nm pixel size) and 20 eV, respectively.

### Structured illumination microscopy.

Isolated *Chlamydomonas* centrioles and procentrioles (also known as basal bodies and probasal bodies, respectively) were centrifuged at 10,000g (JS13.1, Beckman Coulter) onto a 12 mm coverslip (Roth, reference YX03.1) in a 15 ml Corex tube with an adaptor. Coverslips were then fixed for 7 min in –20°C methanol, washed in PBS, incubated 60 min at room temperature with primary antibodies in 1% bovine serum albumin and 0.05% Triton X-100, washed 15 min in PBS and incubated 45 min at room temperature with secondary antibodies. Primary antibodies were 1:300 rabbit CrSAS-6 (ref. 30), 1:200 rabbit Bld10p[16] (gift from Masafumi Hirono) and 1:1,000 mouse glutamylated tubulin (GT335, Adipogen) for Fig. 2d–f. Primary antibodies were 1:1,000 mouse 6xHis (Sigma) and 1:200 rabbit Bld10p for Supplementary Fig. 3b,c. Secondary antibodies were 1:1,000 goat anti-rabbit coupled to Alexa 568 and 1:1,000 goat anti-mouse coupled to Alexa 488. Imaging was carried out on a 3D NSIM Nikon microscope as described[30].

### Linear regression analysis.

Preferential growth in 3D implies that one side of the cartwheel lattice should exhibit more variability than the other, owing to the preferred addition of SAS-6 rings. Therefore, whereas the side with less variability should appear as a plane, that with preferred ring addition should appear as a non-planar object or cloud. To test this possibility, we extracted from four tomograms the coordinates of bottom and top extremities of all cartwheels. Each pair of bottom and top points were defined depending on their $z$ coordinates (the bottom point being defined as having the lower $z$ while the top one having the higher $z$) and constituted two populations of points. We next applied multiple linear regression, using $x$ and $y$ coordinates as predictors for the $z$ coordinate. In 3D, the model returned by this regression corresponds to a plane. To estimate whether the obtained plane was fitting the experimental values, we used the $R^2$ value, which indicates the proportion of variability explained by the model.

In the case of a model perfectly fitting data, $R^2$ equals 1 whereas a less fitting model will have a lower $R^2$ value. All analyses were carried out using R v3.2.3.(https://www.R-project.org/).

### Modelling of cartwheel height.

To develop an effective fit for the measured height distribution, we assumed two independent distributions of multiples of single and double rings of height $h_1$ and $h_2$, respectively. Each distribution is modelled with a Gaussian distribution ($G_1$ and $G_2$), with averages $\mu_1$ and $\mu_2$, respectively. Furthermore, these distributions are truncated to exclude negative heights. To allow only integral multiples of heights $h_1$ and $h_2$, $G_1$ and $G_2$ are then discretized by multiplication for a comb function of periods $h_1$ and $h_2$, respectively. The two resulting individual functions are:

$$G'_1(x_1, \mu_1, \sigma_1, h_1) = G_1(x_1, \mu_1, \sigma_1) \cdot \Xi(h_1)$$
$$G'_2(x_2, \mu_2, \sigma_2, h_2) = G_2(x_2, \mu_2, \sigma_2) \cdot \Xi(h_2)$$

We then considered the overall cartwheel height $X$ as the sum of the contributions of single ($X_1$) and double ($X_2$) rings. Therefore, the distribution of $X = X_1 + X_2$, where $X_1$ is a variable with a Gaussian distribution $G'_1 (X_1 \sim G'_1)$ and, likewise, $X_2 \sim G'_2$ reads as:

$$P(X = x) = \sum_{k=0}^{\infty} G'_1(X_1 = k) G'_2(X_2 = x - k)$$

It is then convenient to introduce the new variable $\mu$, the sum of the two averages of the individual distributions, as well as $R$, the relative contribution to cartwheel height of the second species:

$$\mu = \mu_1 + \mu_2$$
$$R = \frac{\mu_2}{\mu_1 + \mu_2}$$

Therefore, $\mu_1$ and $\mu_2$ can be expressed as:

$$\mu_1 = (1 - R)\mu$$
$$\mu_2 = (R)\mu$$

In this way, $\mu_1$ and $\mu_2$ represent the individual contributions of each species to the average cartwheel height. By definition, the resulting distribution can be written then as the convolution of the individual distributions:

$$P(x, \mu, R, \sigma_1, \sigma_2, h_1, h_2) = \left\{ G'_1[x, (1 - R)\mu, \sigma_1, h_1] \right\} \otimes \left\{ G'_2[x, R \cdot \mu, \sigma_2, h_2] \right\}$$

To model the data, one needs to also take into account experimental uncertainties; to accomplish this, the above distribution is further convolved with a Gaussian function $G_{exp}$, leading to:

$$H(x, \mu, R, \sigma_1, \sigma_2, h_1, h_2, \sigma_{exp}) = P(x, \mu, R, \sigma_1, \sigma_2, h_1, h_2) \otimes G_{exp}(x, 0, \sigma_{exp})$$

Finally, an offset s is introduced to model the effect of a potential different height for the first or the last ring in the stack. The final resulting fitting distribution is then:

$$H'(x, \mu, R, \sigma_1, \sigma_2, h_1, h_2, \sigma_{exp}, s) = H(x - s, \mu, R, \sigma_1, \sigma_2, h_1, h_2, \sigma_{exp})$$

Where $\sigma_{exp}$ was determined experimentally from two independent measurements of the height of 60 cartwheels as $\sigma = \frac{\sqrt{\pi}}{2}\langle|x_1 - x_2|\rangle$, with $x_1$ and $x_2$ being two height measurements of the same cartwheel.

Initially, $\chi^2$ minimization fitting of the distribution of the experimentally measured heights $x_i$ ($N = 1,039$) was performed in IgorPro 7 (Wavemetrics), with $\mu$, $\sigma_1$ and $\sigma_2$ left as free parameters. The fitting was performed for $R \in ]0,1]$ (step size 0.04), $s \in [0,5]$ (step size 1), $h_1 \in [3,11]$ (step size 0.2) and $h_2 \in [12,20]$ (step size 0.2) on the smoothed experimental distribution (running average with window $M = 4$). The fitting parameters $\mu (R, s, h_1, h_2)$, $\sigma_1 (R, s, h_1, h_2)$ and $\sigma_2 (R, s, h_1, h_2)$ thus obtained were then used to calculate the log likelihood function for the same parameter interval as:

$$\log(L(R, s, h_1, h_2)) = \sum_i H'(x_i, \theta)$$

where $\theta$ is the complete set of parameters ($\mu$, $R$, $\sigma_1$, $\sigma_2$, $h_1$, $h_2$, $s$) and $x_i$ the experimentally measured values for cartwheel height. The AIC was then computed as $AIC = -\log(L) + k$, with $k = 7$ being the number of free parameters. Since just one model was fitted with a constant number of parameters, the use of AIC is actually superfluous. Nevertheless, we decided to use this notation to facilitate immediate comparison with models corresponding to uniform populations of single or double rings (namely $R = 0$ and $R = 1$, respectively), which would have a $k = 5$. In addition, the use of AIC will facilitate comparisons between the model proposed here and potential other ones that may be developed in future. $\Delta AIC$ reported in Fig. 5 were then calculated by subtracting the global minimum found for the AIC to set the best model ($R = 0.85$, $s = 3$, $h_1 = 7.8$, $h_2 = 17.6$) as $\Delta AIC = 0$. Note that the obtained value of $R = 0.85$ corresponds to the ratio of average height contribution of the $h_2$ species, which given the height ratio of $h_2/h_1 = 2.25$ corresponds to a number ratio of single ring units/double ring unit $N_2/N_1 = 2.56$. To verify that the results were not influenced by the data smoothing applied for the first part of the fitting, $\Delta AICs$ were calculated for $M \in [1,4]$ in an analogous manner for parameter ranges $R \in ]0,1]$, $h_1 \in [3,11]$ (step size 0.2), $h_2 \in [12,20]$ (step size 0.2) and the resulting $\Delta AIC$ used to generate Supplementary

Fig. 7, which shows that the fitting results hold irrespective of the smoothing applied to the experimental distribution.

**Data availability.** The authors declare that all data supporting the findings of this study are available within the paper and its Supplementary Information files.

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

## Acknowledgements

We thank M. Hirono for sharing reagents and discussing unpublished data; A. De Simone, G. Hatzopoulos and I. Vakonakis for reading the manuscript. Pa.G. and V.H. are currently supported by the Swiss National Science Foundation (SNSF) (grant PP00P3_157517, which also supports M.L.G.), I.I. and B.Z. by a professorship grant from the SNSF PP00P3_139098 and PP00P3_163761 to B.Z. Some of the imaging was performed with a device supported by the Microscopy Imaging Center (MIC) of the University of Bern (FEI Tecnai F20). This work was also supported by grant AdG 340227 from the European Research Council to P.G, which funded Pa.G., V.H., V.N. and I.F.

## Author contributions

Pa.G. and V.H. designed, performed and analysed all the experiments of the paper and wrote the manuscript. M.L.G. analysed the data in Figs 3,4 and 5c and Supplementary Figs 3g,4a and 8. N.B. designed the mathematical model in Fig. 5a,b and Supplementary Fig. 7 and wrote the manuscript. I.I. and B.Z. provided access to the EM microscope F20 and helped with data acquisition in Figs 3 and 4. I.F. purified the recombinant proteins presented in Supplementary Fig. 1c–e. V.N. performed EM-negative staining experiment in Fig. 3f and Supplementary Fig. 4b. K.N.G., H.S. and D.L. provided electron microscopy support and discussed the results. P.G. designed, analysed and supervised the work and wrote the manuscript.

## Additional information

**Competing interests:** The authors declare no competing financial interests.

