## [Peer Review File · Nature Communications]

REVIEWERS' COMMENTS:

Reviewer #1 (Remarks to the Author):

The authors have substantially improved the modeling component of the manuscript based on the points raised by myself and the other reviewers. They have consolidated to a single model, which has been analyzed in a systematic way that in general supports their conclusions about the relative importance of single and double ring addition in cartwheel growth. I still have a couple of concerns regarding both presentation and analysis that I think should be addressed prior to publication:

1) Presentation: I apologize for not raising this concern in the initial review, but the Gaussian model for cartwheel height has become the sole model in the revised manuscript, so I think it's important for the model to be clearly understood and reproducible. The description of the model on pp. 19-20 in terms of convolutions of Gaussians multiplied by comb functions leading to the formula between lines 474 and 475 is a bit esoteric and will not be familiar to the general reader. I struggled for quite a while to figure out the mathematical form of the function being described. I think the current description is a bit loose in terms of describing the actual functions of x that make up the fitting distribution, but I admit to not being completely comfortable with the notation being used. The formula would be more understandable to me as a sum over cartwheels of composition n_1 , n_2 , where n_1 is the number of single rings and n_2 is the number of double rings. Each term in this sum contributes a Gaussian in x centered at $n_1 \cdot h_1 + n_2 \cdot h_2$ with width σ_{exp} , the experimentally determined variability. The probability of each such term is computed as a product of a single ring and double ring probabilities, each of which is assumed to be an independent Gaussian distribution, etc. etc.

2) Analysis: I am not sure that what the authors are calling a log likelihood function is in fact the correct log likelihood for this model given the data described. I mentioned this concern in my previous review in my comments about their χ^2 function. The problem is that the function they are fitting to already incorporates the experimental error, so it's not clear how to determine the probabilities associated with deviations between the model and the experimental data, which is now computed with rolling averages. I'm not an expert in how to do this, but I think it could be done by splitting the data in half and averaging the differences between the data points. I think this noise estimate may depend on the smoothing window size. I don't think it's necessary to do all of this, but an easier workaround is to just call the metric being used a "goodness-of-fit" indicator rather than χ^2 or log likelihood, because it's not exactly. A corollary is that referring to the fit metric as the AIC here is also probably not valid - it's also a bit superfluous since the AIC (as opposed to just the log likelihood) is required when comparing models with different numbers of parameters, which was true in the original paper but is not the case now, when only a single model is being considered. The AIC adds just a constant value to the fit metric, which then disappears because only the difference is being considered.

The issue that remains then is how to determine when differences in the fit metric are sufficient to justify inferences about the parameters and related mechanisms. For example, looking at Figure 5 it's pretty clear that the model strongly favors a narrow range of h_2 values corresponding to the double ring height, but h_1 seems comparatively less constrained even though it does have an optimum near the mechanistically significant value of 8 nm. I think the authors should note that the h_1 value is not as tightly constrained. The claim for a value of $s \sim 3$ nm seems even more tenuous. Looking at Fig. 5b one sees the orange arrow indicating the position of the minimum in the middle of a long valley that spans the full range of s investigated. The results seem to favor nonzero s but suggests that the fitting is not very sensitive to s . The vertical valley in Supp. Fig. 7b for M is not all that different, and is used to support the claim that the fitting doesn't depend significantly on M . If the authors believe that s is important, it might be worthwhile to show as an additional panel to Supp Fig. 7 or even an additional figure predicted distributions for different values of s . I spent quite a bit of time looking at Supp. Fig. 7 and it's hard to convince myself that

the model is really capturing the essential features of the experimental distribution - i.e. that the model wiggles and the experimental wiggles really coincide. I think it would help to show some bad fits to make this point clearer to readers.

Reviewer #2 (Remarks to the Author):

In their revised manuscript Guichard et al. have done a nice job of addressing my major concerns. This is an important piece of work and the data is striking and is generally well presented. I strongly support publication in Nature Communications and have only two very minor points that the authors might like to consider prior to publication.

1. The schematic diagram in Figure S2a does not seem to be well aligned with what the pictures show. Section 2 (microtubules + cartwheel) seems to illustrate a mix of two zones: microtubules + cartwheel and microtubules alone. Also, I think Section 3 should include a section where one sees the cartwheel just associated with the A-microtubule, but this is not illustrated well on the schematic.

2. It struck me that the EM image of the structures formed by CrSas-6_FL and Bld10p_C (Figure 2g and Figure S3d) bear a remarkable resemblance to the structures formed when Sas-6 and Ana2/STIL are overexpressed in fly cells (Stevens et al., Dev. Cell, 2010). This might be worth commenting on as this would suggest that similar structures can be formed in vivo in a different system. More importantly, it suggests that these structures are likely primarily formed of Sas-6 and Bld10 and that while Ana2/STIL is important for the formation of these structures in fly cells, it is perhaps unlikely to be an intrinsic component of the structure.

Reviewer #4 (Remarks to the Author):

The authors have done an outstanding job of addressing the concerns expressed in the original reviews. This will be an important contribution to the field.

Response to Referees' comments

We thank the three reviewers for their assessment of our manuscript and for being supportive of publication. We explain below how we addressed the minor outstanding comments expressed by Reviewers #1 and #2.

Reviewer #1

The authors have substantially improved the modeling component of the manuscript based on the points raised by myself and the other reviewers. They have consolidated to a single model, which has been analyzed in a systematic way that in general supports their conclusions about the relative importance of single and double ring addition in cartwheel growth. I still have a couple of concerns regarding both presentation and analysis that I think should be addressed prior to publication:

1) Presentation: I apologize for not raising this concern in the initial review, but the Gaussian model for cartwheel height has become the sole model in the revised manuscript, so I think it's important for the model to be clearly understood and reproducible. The description of the model on pp. 19-20 in terms of convolutions of Gaussians multiplied by comb functions leading to the formula between lines 474 and 475 is a bit esoteric and will not be familiar to the general reader. I struggled for quite a while to figure out the mathematical form of the function being described. I think the current description is a bit loose in terms of describing the actual functions of x that make up the fitting distribution, but I admit to not being completely comfortable with the notation being used. The formula would be more understandable to me as a sum over cartwheels of composition n_1, n_2 , where n_1 is the number of single rings and n_2 is the number of double rings. Each term in this sum contributes a Gaussian in x centered at $n_1 \cdot h_1 + n_2 \cdot h_2$ with width σ_{exp} , the experimentally determined variability. The probability of each such term is computed as a product of a single ring and double ring probabilities, each of which is assumed to be an independent Gaussian distribution, etc. etc.

> We thank the reviewer for these valuable suggestions, and apologize for not having been sufficiently clear in our writing of the previous version of the manuscript. To address this issue, we have significantly expanded and clarified the Methods section to explain in a stepwise fashion how the model was developed (pages 20-23 of the revised manuscript).

2) Analysis: I am not sure that what the authors are calling a log likelihood function is in fact the correct log likelihood for this model given the data described. I mentioned this concern in my previous review in my comments about their χ^2 function. The problem is that the function they are fitting to already incorporates the experimental error, so it's not clear how to determine the probabilities associated with deviations between the model and the experimental data, which is now computed with rolling averages. I'm not an expert in how to do this, but I think it could be done by splitting the data in half and averaging the differences between the data points. I think this noise estimate may depend on the smoothing window size. I don't think it's necessary to do all of this, but an easier workaround is to just call the metric being used a "goodness-of-fit" indicator rather than χ^2 or log likelihood, because it's not exactly.

> The concern expressed by the Reviewer probably stems from us being insufficiently explicit about the fitting methodology in the previous version of the manuscript. Just to be clear, the reported log likelihood was not calculated from the experimentally determined distribution with moving average, but instead from the discrete measured heights. Therefore, although the log likelihood for the model includes the experimentally measured error, this pertains to the height of individual cartwheels and not to the error on the distribution. These points are stated more clearly in the Methods section of the revised manuscript (page 21).

A corollary is that referring to the fit metric as the AIC here is also probably not valid - it's also a bit superfluous since the AIC (as opposed to just the log likelihood) is required when comparing models with different numbers of parameters, which was true in the original paper but is not the case now, when only a single model is being considered. The AIC adds just a constant value to the fit metric, which then disappears because only the difference is being considered.

> We agree with the Reviewer that AIC is somehow superfluous in this case since only one model with a fixed number of degrees of freedom is considered. Nevertheless, we decided to retain the Δ AIC notation, because this allows for an immediate comparison with models corresponding to uniform populations of single or double rings (namely $R=0$ and $R=1$, respectively), which have less degree of freedom. In addition, the use of AIC will facilitate comparisons between the model developed here and other ones that may be proposed in the future. These points are stated explicitly on pages 21-23 of the revised manuscript.

The issue that remains then is how to determine when differences in the fit metric are sufficient to justify inferences about the parameters and related mechanisms. For example, looking at Figure 5 it's pretty clear that the model strongly favors a narrow range of h_2 values corresponding to the double ring height, but h_1 seems comparatively less constrained even though it does have an optimum near the mechanistically significant value of 8 nm. I think the authors should note that the h_1 value is not as tightly constrained. The claim for a value of $s \sim 3$ nm seems even more tenuous. Looking at Fig. 5b one sees the orange arrow indicating the position of the minimum in the middle of a long valley that spans the full range of s investigated. The results seem to favor nonzero s but suggests that the fitting is not very sensitive to s . The vertical valley in Supp. Fig. 7b for M is not all that different, and is used to support the claim that the fitting doesn't depend significantly on M . If the authors believe that s is important, it might be worthwhile to show as an additional panel to Supp Fig. 7 or even an additional figure predicted distributions for different values of s .

> We thank the reviewer for suggesting that we discuss further the implications of the fitting process. As a result, we now explain on page 9 (Results section) and on page 29 (legend of Figure 5) that the value of h_1 is less constrained than that of h_2 , and that the value of s is weakly constrained.

I spent quite a bit of time looking at Supp. Fig. 7 and it's hard to convince myself that the model is really capturing the essential features of the experimental distribution - i.e. that the model wiggles and the experimental wiggles really coincide. I think it would help to show some bad fits to make this point clearer to readers.

> As mentioned previously, we have explored several possible models, and found that the one reported in the present version of the manuscript best fits the experimental data. Given that there are already 8 Supplementary figures, and given also that we do not claim that the model proposed here is a perfect representation of reality (something we now spell out more explicitly on page 10 of the revised manuscript), we are of the opinion that showing bad fits would not be helpful.

Reviewer #2

In their revised manuscript Guichard et al. have done a nice job of addressing my major concerns. This is an important piece of work and the data is striking and is generally well presented. I strongly support publication in Nature Communications and have only two very minor points that the authors might like to consider prior to publication.

1. The schematic diagram in Figure S2a does not seem to be well aligned with what the pictures show. Section 2 (microtubules + cartwheel) seems to illustrate a mix of two zones:

microtubules + cartwheel and microtubules alone. Also, I think Section 3 should include a section where one sees the cartwheel just associated with the A-microtubule, but this is not illustrated well on the schematic.

> We thank the Reviewer for this useful comment; we have incorporated her/his suggestions when revising Supplementary Fig. S2a.

2. It struck me that the EM image of the structures formed by CrSas-6_FL and Bld10p_C (Figure 2g and Figure S3d) bear a remarkable resemblance to the structures formed when Sas-6 and Ana2/STIL are overexpressed in fly cells (Stevens et al., Dev. Cell, 2010). This might be worth commenting on as this would suggest that similar structures can be formed in vivo in a different system. More importantly, it suggests that these structures are likely primarily formed of Sas-6 and Bld10 and that while Ana2/STIL is important for the formation of these structures in fly cells, it is perhaps unlikely to be an intrinsic component of the structure. nt of the structure.

> We thank the Reviewer for noticing this interesting parallel. We included this point in a new paragraph in the Discussion section of the revised manuscript (page 11).